# GenNet framework: interpretable deep learning for predicting phenotypes from genetic data

Arno van Hilten [1✉], Steven A. Kushner [2], Manfred Kayser [3], M. Arfan Ikram [4], Hieab H. H. Adams [1,5], Caroline C. W. Klaver [4,6], Wiro J. Niessen[1,7,8] & Gennady V. Roshchupkin [1,4,8✉]

Applying deep learning in population genomics is challenging because of computational issues and lack of interpretable models. Here, we propose GenNet, a novel open-source deep learning framework for predicting phenotypes from genetic variants. In this framework, interpretable and memory-efficient neural network architectures are constructed by embedding biologically knowledge from public databases, resulting in neural networks that contain only biologically plausible connections. We applied the framework to seventeen phenotypes and found well-replicated genes such as *HERC2* and *OCA2* for hair and eye color, and novel genes such as *ZNF773* and *PCNT* for schizophrenia. Additionally, the framework identified ubiquitin mediated proteolysis, endocrine system and viral infectious diseases as most predictive biological pathways for schizophrenia. GenNet is a freely available, end-to-end deep learning framework that allows researchers to develop and use interpretable neural networks to obtain novel insights into the genetic architecture of complex traits and diseases.

[1] Department of Radiology and Nuclear Medicine, Erasmus MC, Medical Center, Rotterdam, the Netherlands. [2] Department of Psychiatry, Erasmus MC, Medical Center, Rotterdam, the Netherlands. [3] Department of Genetic Identification, Erasmus MC, Medical Center, Rotterdam, the Netherlands. [4] Department of Epidemiology, Erasmus MC, Medical Center, Rotterdam, the Netherlands. [5] Department of Clinical Genetics, Erasmus MC, Medical Center, Rotterdam, the Netherlands. [6] Department of Ophthalmology, Erasmus MC, Medical Center, Rotterdam, the Netherlands. [7] Faculty of Applied Sciences, TU Delft, Delft, the Netherlands. [8] These authors jointly supervised this work: Wiro J. Niessen and Gennady V. Roshchupkin. ✉email: a.vanhilten@erasmusmc.nl; g.roshchupkin@erasmusmc.nl

Wh ile genome-wide association studies (GWAS) have identified numerous genomic loci associated with complex traits and diseases, the biological interpretation of the underlying mechanisms often remains unclear. Recent GWAS studies with increasingly large sample sizes are resulting greater numbers of significant associations, at an increasing number of independent loci. To illustrate, the latest GWAS for body height based on 700,000 individuals identified more than 3000 near-independent significantly associated single nucleotide polymorphisms (SNPs)[1]. Uncovering a clear biological interpretation from all this information is a challenging task, in which causal variants, genes, and pathways need to be identified. In response, many methods such as MAGMA[2], ALIGATOR[3], and INRICH[4], have been developed to obtain a biological interpretation from GWAS summary statistics, providing insights into relevant genes and pathways for defined phenotypes of interest. These methods explore GWAS summary statics and utilize knowledge from annotated biological databases such as NCBI RefSeq[5], KEGG[6], Reactome[7], and GTEx[8], which have proven to contain crucial information for understanding the underlying biological mechanisms of the human genome[9]. Additionally, it has been shown that embedding biological knowledge from these databases in polygenic risk scores can improve interpretation, trans-ancestry portability, and genetic risk prediction[10–12].

Given the increasing amount of data available via biobanks and new developments to integrate data, it is now feasible to analyze raw data with more advanced methods. Deep learning is the state of the art in many domains such as medical image analysis[13] and natural language processing[14] because of its flexibility and modeling capabilities. In many cases, deep learning yields better performance than traditional approaches, since it scales very well with data size and can model highly non-linear relationships. However, a limitation to deep learning is that these algorithms are often uninterpretable because of their complexity[15,16]. Additionally, genetic data does not lend itself well to the convolution operation, the main driver of the success of deep learning in the imaging domain. Traditional fully connected neural networks have been successfully applied to predict genetic risk. Recently, Badre et al.[17] employed a fully connected neural network for improving polygenic risk scores for breast cancer, training a neural network with up to 528,620 input variants. However, these networks are very memory-intensive and therefore often require pre-selecting SNPs using GWAS summary statistics. Applying these fully connected neural networks for millions of input variants would require infeasible amounts of computational time and memory.

To overcome these limitations, we propose GenNet, a novel framework for predicting phenotype from genotype. Within the GenNet framework, biological information from annotated biological sources such as NCBI RefSeq, KEGG, and single RNA gene expression datasets, is used to define biologically plausible connections. As a result, neural networks based on this framework are memory efficient, interpretable, and yield biological interpretations for their predictions. GenNet is an end-to-end deep learning framework available as a command-line tool (https://github.com/ArnovanHilten/GenNet/).

## Results

### A framework for constructing interpretable neural networks for phenotype prediction. The main concept of the GenNet framework is summarized graphically in Fig. 1. In this framework, prior knowledge is used to create groups of connected nodes to reduce the number of learnable parameters in comparison to a fully connected neural network. For example, in the first layer, biological knowledge in the form of gene annotations

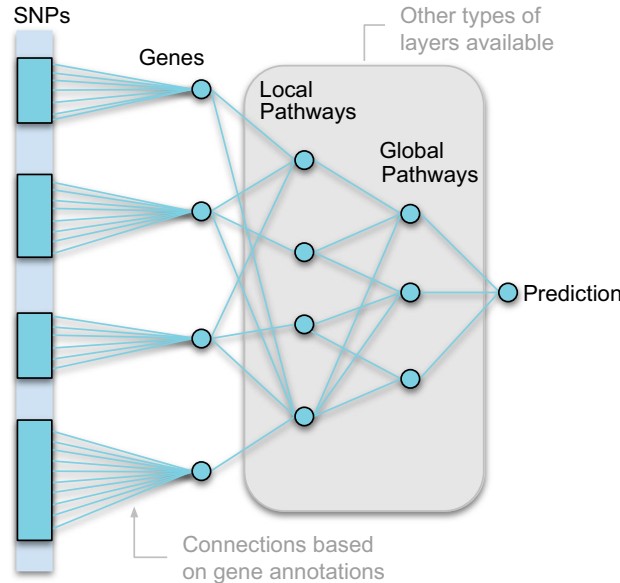

**Fig. 1 Overview of the GenNet framework.** Neural networks are created by using prior biological knowledge to define connections between layers (i.e., SNPs are connected to their corresponding genes by using gene annotations, and genes are connected to their corresponding pathway by using pathway annotations). Prior knowledge is thus used to define each connection, creating interpretable networks.

is used to group millions of SNPs and to connect those SNPs to their corresponding genes. The resulting layer retains only meaningful connections, significantly reducing the total number of parameters compared to a classical fully connected layer. Because of this memory-efficient approach networks in the GenNet framework are able to handle millions of inputs for genotype-to-phenotype prediction.

Biological knowledge is thus used to define only meaningful connections, shaping the architecture of the neural network. Interpretability is inherent to the neural network's architecture; each node is uniquely defined by its connections and represents a biological entity (e.g., gene, pathway). For example, a network that connects SNPs-to-genes and genes-to-output. The learned weights of the connections between layers represent the effect of the SNP on the gene or the effect of the gene on the output. In the network, all neurons represent biological entities, and weights model the effects between these entities, together forming a biologically interpretable neural network.

Many types of layers can be created using this principle. Apart from gene annotations, our framework provides layers built from exon annotations, pathway annotations, chromosome annotations, and cell and tissue type expressions. All these layers can be used like building blocks to form new neural network architectures.

We built a proof of concept simulations as shown in Fig. 2a and described in Supplementary Method 1. These simulations demonstrate that the proposed network architecture is interpretable, the strongest weights are assigned to causal variants and genes. Next, we designed experiments to test the network architecture's performance under a variety of conditions. Performance is dependent on phenotype, neural network architecture, and dataset size. To disentangle this, we created simulated data with varying levels of heritability, the number of training samples, and polygenicity (Supplementary Method 1). Figures 2b, c demonstrate the major trends observed in the simulations. As expected, the network performs best for traits

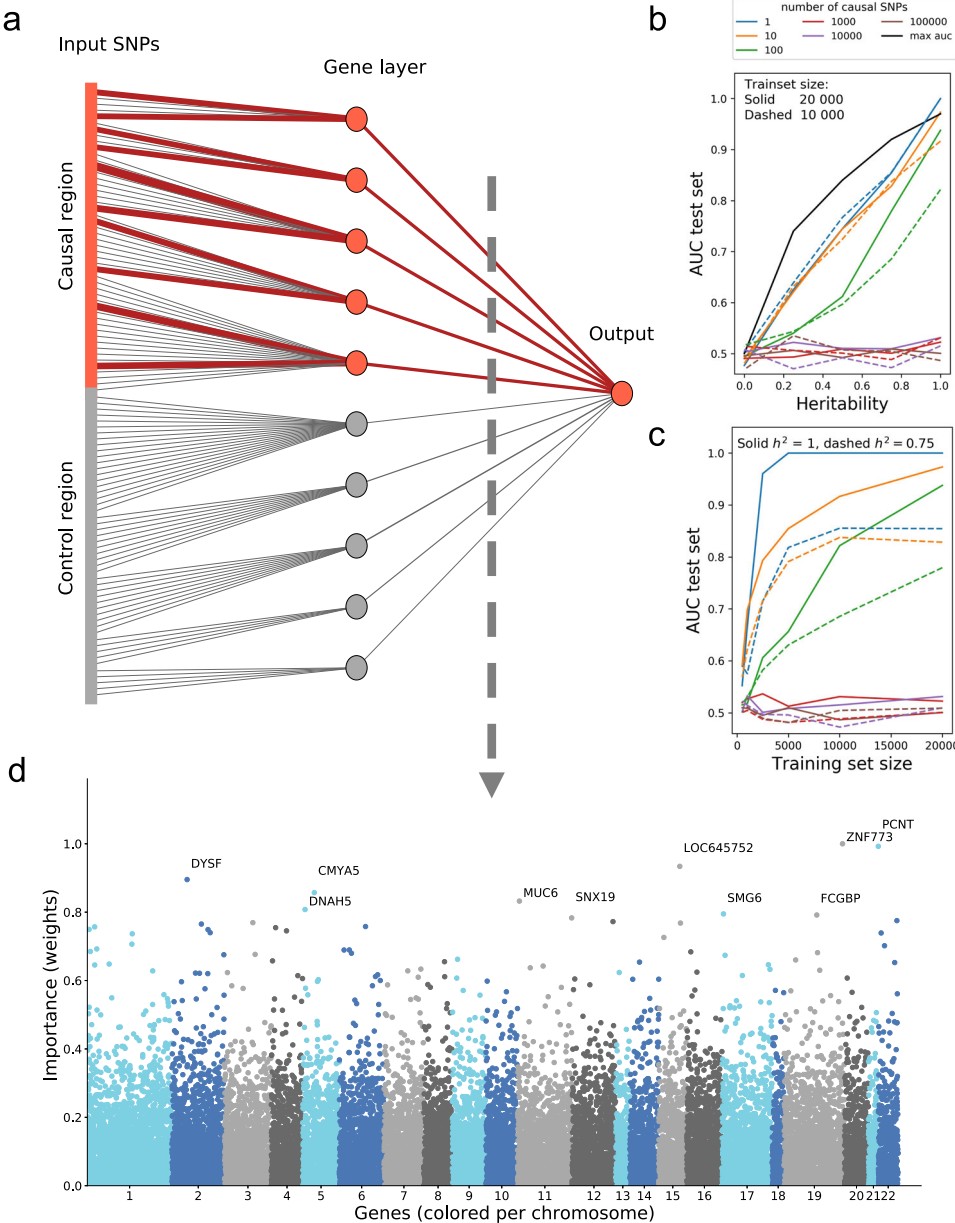

**Fig. 2 Simulation results and the importance of each gene for predicting schizophrenia. a** A simple, non-linear proof of concept. In this simulation, each gene in the causal region has two causal SNPs that cause the simulated disease. The magnitude of the learned weight is represented by the line thickness (contributing causal connections in red, non-contributing control connections in gray). **b** A secondary set of simulations show the performance of GenNet, expressed as area under the curve, for increasing levels of heritability and training set size (**c**). The black curve presents the theoretical maximum of the AUC versus heritability. **d** Manhattan plot showing genes and their relative importance according to the network, here we have shown the results for distinguishing between schizophrenia cases and controls in the Sweden exome study. This Manhattan plot is a cross-section between the gene layer (21,390 nodes) and the outcome of a trained network with 1,288,701 input variants.

with high heritability, high number of training samples, and low polygenicity.

**Applying neural network architectures based on gene-annotations to population-based data.** Motivated by the proof-of-concept and outcomes of the simulations, we applied the framework to data from multiple sources, including population-based data from the UK Biobank study[18], the Rotterdam study[19], and the Swedish Schizophrenia Exome Sequencing study[20]. Phenotypes vary from traits that can be predicted well from only a dozen variants (eye color) to disorders in which thousands of variants explain only a small portion of the variance (schizophrenia and bipolar disorder)[21,22]. The datasets also vary in size

and type of data. We used 113,241 exonic variants from imputed microarray-based GWAS data from the Rotterdam study for predicting eye color while we predict fifteen phenotypes in the UK Biobank using 6,986,636 input variants from whole exome sequencing (WES) data. Finally, we used 1,288,701 WES input variants for predicting schizophrenia in the Swedish study. An overview of the main results for networks embedded with gene and pathway information can be found in Table 1. The results for all the experiments, including expression-based networks, can be found in Supplementary Note 2.

Phenotypes with more training samples and that require less variants to obtain high predictive performance, such as eye and hair color, yielded the best performance. Nonetheless, we

**Table 1 Overview of the main results for the experiments using the Rotterdam Study, UK Biobank, and Swedish Schizophrenia Exome Sequencing study data.**

| Dataset (type) | Trait | Subjects & phenotype Class I | Class II | AUC gene test (val) | Top three predictive genes | AUC pathway test (val) | Top predictive pathway Global level | Middle level | Local level |
|---|---|---|---|---|---|---|---|---|---|
| Rotterdam Study (genotype array) | Eye color | 4041 Blue | 2250 Other | **0.75 (0.74)** | HERC2, OCA2, LAMC1 | 0.50 (0.52) | Organismal Systems (78.4%) | Digestive system (72.6%), | Pancreatic secretion (59.1%) |
| UK Biobank (exome) | Hair color | 1734 Red | 1727 Other | **0.93 (0.94)** | MC1R*, SHOC2, DCTN3 | 0.77 (0.77) | Genetic Information Processing (87.4%) | Replication and repair (83.4%) | Fanconi anemia pathway (79.7%) |
| | | 3762 Black | 3753 Other | 0.80 (0.82) | OCA2, RPL23AP8, MC1R* | 0.76 (0.78) | Organismal Systems (46.9%) | Endocrine system (18.6%) | Axon guidance (5.0%) |
| | | 4501 Blond | 4518 Other | **0.66 (0.65)** | OCA2, TC2N, SLC45A2 | 0.58 (0.57) | Organismal Systems (70.2%) | Endocrine system (30.1%), | Adrenergic signaling in cardiomyocytes (4.1%) |
| | Bipolar disorder | 343 Cases | 347 Controls | **0.60 (0.56)** | LINC00266-1, CSMD1, TCERG1L | 0.47 (0.55) | Organismal Systems (76.9%) | Endocrine system (46.5%) | Melanogenesis (32.9%) |
| | Atrial fibrillation | 192 Cases | 194 Controls | **0.56 (0.59)** | **BRINP1, SORBS3, ELMOD3 | 0.57 (0.63) | Organismal Systems (39.6%) | Signal transduction (11.2%) | Cytokine–cytokine receptor interaction (4.4%) |
| | Coronary Artery Disease | 1563 Cases | 1600 Controls | **0.56 (0.58)** | **STARD7-AS1, VWC2L, NSD2 | 0.54 (0.56) | Environmental Information Processing (29.5%), | Signal transduction (27.7%) | PI3K-Akt signaling pathway (4.6%) |
| | Dementia | 139 Cases | 142 Controls | **0.60 (0.65)** | RPL23AP87, CTNNA3, LINC01003 | 0.55 (0.58) | Human Diseases (39.8%) | Signal transduction (22.2%) | Pathways in cancer (5.6%) |
| | Male balding pattern | 3454 Balding pattern 1 | 3454 Balding pattern 4 | **0.56 (0.57)** | NGEF, NKRD18B, SYNJ2 | 0.54 (0.55) | Organismal Systems (34.6%) | Nervous system (9.7%) | Metabolic pathways (8.7%) |
| | Asthma | 4229 Cases | 4214 Controls | **0.55 (0.57)** | HLA-DQB1, HCG9, LINC00266-1 | 0.51 (0.54) | Genetic Information Processing (52.3%) | Folding, sorting and degradation (41.5%) | Ubiquitin mediated proteolysis (22.0%) |
| | Diabetes | 2557 Cases | 2555 Controls | **0.54 (0.57)** | **DNAH10, SNAR-I, PSMD13 | 0.54 (0.54) | Environmental Information Processing (43.5%), | Signal transduction (40.5%), | Ras signaling pathway (7.9%) |
| | Breast cancer | 1070 Cases | 1082 Controls | **0.51 (0.56)** | RPL23AP87, LINC00266-1, HPSE | 0.51 (0.56) | Human Diseases (57.1%) | Infectious diseases: Viral (16.6%) | Pathways in cancer (6.5%) |
| Sweden (exome) | Schizophrenia | 4969 Cases | 6245 Controls | **0.74 (0.73)** | ZNF773, PCNT, DYSF | 0.68 (0.67) | Human Diseases (30.8%) | Infectious diseases: Viral (27.3%) | Human papillomavirus infection (11.7%) |

The performance in AUC for the network with gene-annotations is bold if the network outperformed or matched LASSO regression. Manhattan plots for the genes can be found in Supplementary Notes 3–5. *MC1R was not annotated but was identified by linkage disequilibrium. **Many genes contributed to the prediction without clear separation between genes (see Supplementary Note 4).

achieved a good predictive performance for schizophrenia, a highly polygenic disorder, with an area under the curve (AUC) of 0.74 in the held-out test set. All models based on gene annotations outperformed or matched the baseline LASSO logistic regression model with the exception of the black hair color phenotype.

Inspecting the networks, we found that the *OCA2* gene was the most important gene to predict blond, dark, and light brown hair color. *OCA2* is involved in the transport of tyrosine, a precursor of melanin[23]. The *OCA2* signal is likely amplified by the nearby gene *HERC2*, previously identified via functional genetic studies as harboring a strong, long-distance enhancer regulating *OCA2* gene expression underlaying pigmentation variation[23]. According to the network, *OCA2* and *HERC2* are the two most predictive genes for predicting blue iris color. Both genes were previously identified through GWAS studies of hair and eye color [24–26].

In the experiments with schizophrenia as outcome, the network was able to classify cases and controls with a maximum accuracy of 0.685 (mean of 0.663 ± 0.014 over 10 runs). Based on all genetic aspects, we estimate the theoretical upper limit for classification accuracy to be between 0.73 and 0.83 for schizophrenia in this dataset (Supplementary Method 6). The model achieved a maximum area under the receiver operating curve of 0.738 (mean of 0.715 ± 0.016) in the held-out test set over 10 runs, thereby considerably outperforming the LASSO logistic regression baseline which obtained a maximum AUC of 0.649 (mean of 0.644 ± 0.003). The genes *ZNF773, PCNT,* and *DYSF* were assigned the highest weights by the neural network and were thus considered to be the most predictive genes for schizophrenia. However, the polygenic nature of schizophrenia can be seen when comparing the Manhattan plot of the genes (Fig. 2d) to other phenotypes in this study (Supplementary Notes 3−5). More genes contributed to the prediction for schizophrenia than for the other phenotypes examined. As a consequence, the predictive performance deteriorates slower for schizophrenia than for eye color prediction when connections to the most predictive genes are removed before training (see Supplementary Discussion 7).

To evaluate if embedding prior knowledge also improves predictive performance, we compared networks built using gene annotations to networks with the same number of connections but randomly connected. For predicting blue eye color, the network build using gene annotations performed significantly better than a randomly connected neural network over ten runs ($p = 6.5 \times 10^{-3}$). However, for predicting schizophrenia we found the opposite, a randomly connected neural network performed significantly better than networks embedded with prior knowledge ($p = 2.5 \times 10^{-4}$). See Supplementary Discussion 8 for more information on all phenotypes.

**Embedding pathway and expression-based annotations into the neural network architecture**. Figure 3 shows the relative importance of the pathways for predicting schizophrenia. The neural network, embedded with the KEGG pathway information, obtained an AUC of 0.68 in the test set. Human diseases (30.8%), organismal systems (26.7%), and genetic information processing (26.5%) were the main contributors to the neural network's prediction of schizophrenia. The contribution of human diseases was mainly driven by viral infectious diseases (27.3%) which can be subdivided further into human papillomavirus infection (11.7%), herpes simplex infections (3.7%), and human cytomegalovirus infection (3.2%), as well as genes and SNPs assigned to these diseases. The ubiquitin-mediated proteolysis pathway, a subset of the genetic information pathway, had a relative importance of 10.0%.

To investigate if the most predictive genes are concentrated in a single cell or tissue type, we used three different gene expression datasets processed by Finucane et al. (2018)[27] to create layers subsequent to the gene-based layer. The relative importance of the trained networks showed a widespread signal with small contributions by many cell and tissue types. When GTEx data was embedded in the network, uterine genes had the greatest relative importance (4.2%). For GTEx brain expression data, the frontal cortex (15.1%) had the strongest contribution, whereas for immune cell types mesenteric lymph nodes (1.23%) contributed most. An overview of the experiments performed, including GTEx and other expression-based networks for other phenotypes, can be found in Supplementary Note 2.

## Discussion

We presented a novel framework for predicting phenotypes from genotype with interpretable neural networks. The proposed neural networks have connections defined by prior biological knowledge only, reducing the number of connections and therefore the number of trainable parameters. Consequently, the networks can be trained on a single GPU and offer a biological interpretation for their predictions.

In the first set of experiments, simulations showed the network's performance when varying the degree of heritability, polygenicity, and sample size. In the second set of experiments, the framework was applied to UK Biobank, Rotterdam study, and Swedish Schizophrenia Exome Sequencing study data. In these experiments, phenotypes with widely different heritability, sample size, and polygenicity were predicted with generally high predictive performance. For example, we obtained an AUC of 0.96 for red hair color, an AUC of 0.74 for blue eye color, and an AUC of 0.74 for predicting schizophrenia. For schizophrenia, polygenic risk scores from genome-wide association studies obtained a similar AUC of 0.72 in an independent test cohort and a mean AUC of 0.70 in 40 leave-one-out GWAS analyses (AUC ranging from 0.49 to 0.81)[21]. However, a direct comparison between traditional polygenic risk scores and predictions given by neural networks from the GenNet framework would be unfair. The neural networks employed in this study use WES data or variants from the coding regions, while polygenic risk scores generally use all genotype data. Additionally, polygenic risk scores are built based on GWAS data with large sample sizes. For example for schizophrenia, the network was trained with roughly 4,696 cases and 6,969 controls while polygenic risk scores were derived using 32,838 cases and 44,357 controls[21].

Interpretation of the trained networks revealed well-replicated genes for traits with a known etiology such as *HERC2* and *OCA2* for eye color and *OCA2* and *TC2N* for hair color[24–26,28]. For schizophrenia, a disorder with an unclear etiology, the network identified novel genes, including *ZNF773* and *PCNT*. In previous studies, *PCNT* was only nominally associated with schizophrenia, although it is known to have interactions with *DISC1* (Disrupted in Schizophrenia-1)[29]. There is a strong correlation between (prenatal) viral infections and increased risk for developing schizophrenia[30]. It is therefore interesting that by using the KEGG pathway information to create layers, we identified the viral infectious diseases pathway with its associated pathways, genes, and SNPs to be the most important for predicting schizophrenia. The results of these experiments reveal that with GenNet framework can provide insights across multiple functional levels, identifying which SNPs, genes, pathways, and tissue types are important for prediction.

In a comparison between randomly connected neural networks and networks built using gene annotations, we found that embedding gene annotations in the network architecture resulted

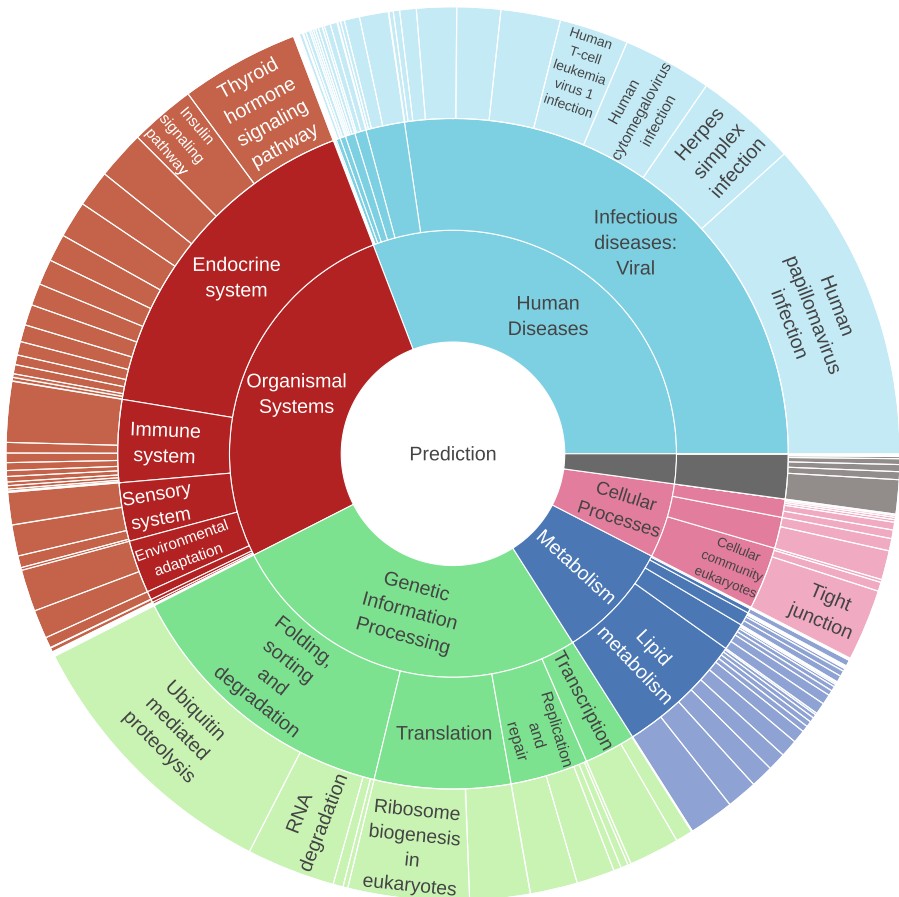

**Fig. 3 Sunburst plot of the important KEGG pathways for predicting schizophrenia.** A neural network with layers based on KEGG pathway information was trained to predict schizophrenia. The relative importance was calculated for each pathway using the learned weights of this neural network. The sunburst plot is read from the center, which shows the output node with the prediction of the neural network. The inner ring represents the last layer with the highest-level pathways, followed by the mid-level pathways, and finally the lowest-level pathways. The gene and SNP layer are omitted for clarity.

in significantly better predictive performance for predicting eye color. However, randomly connected neural networks, with the same number of trainable parameters, performed significantly better for predicting schizophrenia. Based on these experiments using gene annotations, it seems that the optimal neural network architecture depends on the genetic architecture of the trait. But there are many factors that could influence the convergence and thus the predictive performance of a neural network aside from the genetic architecture of the trait. Among others, the quality of the available prior knowledge will be an important factor to consider.

In this study, WES data and exonic variants from microarrays have been used; however, the principles in the GenNet framework can be extended to handle diverse types of input, including genotype, gene expression, and methylation data or combinations thereof. Similarly, we explored only a fraction of the possible layers as building blocks for our networks. Any data that create unique biological groups can be used in the framework to create new layers. However, the quality of the resulting network is highly dependent on the quality of the data used. For example, networks built with biological pathway data performed generally worse than networks using only gene annotations since many genes are not annotated to pathways. But with new and better data from projects such as GTEx[8], ENCODE[31], and KEGG[6] the quality of the layers will only improve. Moreover, given the increasing sample sizes of biobanks in coming years and the development of algorithms, which allow distributed deep learning between cohorts without sharing raw data[32], we foresee that our framework can be easily

adopted for such settings and can be widely used for quantitate traits and functional mapping analysis.

In conclusion, we developed a freely-available framework, which can be used to build interpretable neural networks for genotype data by incorporating prior biological knowledge. In addition to computationally efficient, the architectures are interpretable, thereby alleviating one of the most important shortcomings of neural networks. We have demonstrated the effectiveness of this novel framework across multiple datasets and for multiple phenotypes. Given that each network node is interpretable, we anticipate this approach to have wide applicability for uncovering novel insights into the genetic architecture of complex traits and diseases.

## Methods

**Sweden Schizophrenia Exome Sequencing study**. Sweden-Schizophrenia Population-Based Case-Control Exome Sequencing study (dbGaP phs000473.v2.p2), is a case-control study with 4969 cases and 6245 controls[20]. All individuals aged 18−65, have parents born in Sweden, and provided written informed consent. The following inclusion criteria were used for cases: at least two times hospitalized with schizophrenia discharge diagnosis; no hospital register diagnosis consistent with a medical or other psychiatric disorder that mitigates the schizophrenia diagnosis. Cases were excluded if they had a relationship closer than 2nd degree relative with any other case. Controls were matched to cases and controls were never hospitalized with a discharge diagnosis of schizophrenia. Controls were excluded if they had any relation to either case or control.

The downloaded plink files were converted using HASE[33] to hierarchical data format (HDF), a format that allows fast and parallel data reading. After conversion, the data was transposed and SNPs without any variance were removed (1,288,701

SNPs remain). The data was split into a training, validation, and test set (ratio of 60/20/20), while preserving the ratio cases and controls.

**UK Biobank**. We applied the framework to multiple phenotypes in the UK Biobank using the first release of the WES data, providing WES for 50,000 UK Biobank participants[34]. Phenotypes are self-reported using touchscreen questions in the UK Biobank Assessment Center. Similar to the Sweden cohort all variants without variance were removed, data was converted to HDF and transposed. For every phenotype, an equal number of cases and controls are sampled. The resulting dataset is split into a train, validation, and a test set (ratio of 60/20/20). Related cases and cases with related controls, (kinship > 0.0625) are all in the training set. This is done under the assumption that related cases and controls could ease training, the shared genetic information could steer the network towards the discriminatory features. The validation and test sets contain only unrelated cases and controls within and between sets. Unrelated controls are randomly sampled and added to gain an even distribution between cases and controls in all sets. Misaligned SNPs and sex chromosomes were masked in the first layer and therefore not included in the study.

**Rotterdam Study**. The Rotterdam Study is a prospective population-based cohort study[19]. We used the first cohort consisting of 6291 participants, genotyped using the Illumina 550 and 550 K duo arrays. Samples with low call rate (<97.5%), excess autosomal heterozygosity (>0.336), or sex-mismatch were excluded, as were outliers identified by the identity-by-state clustering analysis (outliers were defined as being >3 standard deviation (SD) from the population mean or having identity-by-state probabilities > 97%). For imputation the Markov Chain Haplotyping (MACH) package version 1.0 software (Imputed to plus strand of NCBI build 37, 1000 Genomes phase I version 3) and minimac version 2012.8.6 were used (call rate > 98%, MAF > 0.001 and Hardy–Weinberg equilibrium $P$-value > $10^{-6}$). From here on, processing steps are identical as described for Sweden Schizophrenia resulting in a dataset with 113,241 exonic variants for 6291 subjects.

For each subject, both eyes were examined by an ophthalmological medical researcher, and eye (iris) color was categorized into three categories; blue, intermediate, and brown using standard images and based on the predominant color and pigmentation[35].

**Prior knowledge**. All SNPs were annotated using Annovar[36]. From these annotations, a sparse connectivity matrix was created, connecting the SNPs to their corresponding genes. Connectivity matrices between SNPs, exons, and genes were made using intron-exon annotations. A mapping between genes and pathways was made using GeneSCF[37] and the KEGG database[6]. Due to the hierarchical structure of the KEGG database no further preprocessing was necessary and the structure could be integrated in GenNet as it is. Expression-based layers were created using publicly available group-wise t-score statistics by Finucane et al. (2018)[27]. For each tissue, we connected the genes with the top 10% highest t-statistic to a tissue to ensures that nodes are uniquely defined by their connections and therefore interpretable. This threshold is in accordance with the work of Finucane et al. (2018)[27], they showed that their results were stable for thresholds of 5, 10, and 20% (URLs can be found in Supplementary Note 9).

Additional to the used layers, there are countless of different network layers possible. Any prior knowledge that groups data uniquely can be used to create layers in the GenNet framework.

**Neural network architecture**. In the GenNet framework, layers are available built from biological knowledge such as exon annotations, gene annotations, pathway annotations, cell expression, and tissue expression. Information from these resources is used to define only meaningful connections, shaping an interpretable and lightweight neural network, allowing the evaluation of millions of input variants together. These networks bear similarities to the first generation of neural networks, where prior knowledge was also used to make neural networks computationally cheaper. Recently, interest for these types of networks has rekindled for biological applications[38–41].

However, the use of prior knowledge also restricts the network layout. The shape of the network, i.e., the number of neurons and the number of layers is determined by the prior knowledge embedded in the network (characteristics of all architectures can be found in Supplementary Note 9). Additionally, for a fair interpretation of the weights, each layer is preceded by batch normalization (without scaling and centering) resulting in an input for each layer with zero mean and unit standard deviation. For all classification task, a sigmoid function was used as a final activation function.

Operating within these restrictions, we optimized the following hyperparameters: the activation function, optimizer, and the L1 kernel regularization penalty. In the first set of experiments to determine the activation function, we found that the hyperbolic tangent activation function consistently led to a higher area under the curve in the validation set than the Relu and PreLu activation functions for schizophrenia, hair color, and eye color. Using the hyperbolic tangent activation function, we optimized the remaining hyperparameters. For each phenotype, we tested different combinations of optimizers with different learning rates and L1 kernel regularization penalties. We

explored combinations of the Adadelta optimizer (with learning rate of 1) or Adam optimizer (learning rates of 0.01, 0.001, and 0.0001) with kernel L1 regularization penalties of 0.1, 0.001, 0.001, 0.00001, and 0 (no penalty). The L1 regularization penalty penalizes the network for the total sum of weights, similar to LASSO regression. A higher penalty will reduce the number of variants the network will use for prediction and is therefore related to the polygenicity of the phenotype.

All models were trained on a single GPU with 11 GB memory (Nvidia GeForce GTX 1080 Ti) using a batch size of 64. The networks were optimized using the ADAM or Adadelta optimizers, using weighted binary cross-entropy with weights proportional to the imbalance of the classes. For regression tasks, mean squared error was used as a loss function in combination with ReLu activations (Supplementary Method 10). The training was stopped after not improving on the validation loss for 10 epochs (all models converged within five days). For each phenotype, the configuration of hyperparameters with the best performance in the validation set was used for a final evaluation in the test set.

**Interpretation**. Interpretation of the network is straightforward due to the simplicity of the concept, the stronger the weight is the more it contributes to the final prediction of the network. The simplest network in the framework, a network built by gene annotations, can be seen as ~20,000 (number of genes) parallel regressions followed by a single logistic regression. The learned weights in these regressions are similar to the coefficients in logistic regression. Especially the last node, the equation for a single neuron with a sigmoid activation (1) is very similar to the equation for logistic regression (2). For both, all inputs ($x_i$ to $x_n$) are multiplied with learnable parameters followed by an addition of a learnable bias $B$ to obtain output $Y$.

$$Y = \text{Sigmoid}\left(\sum_{i=0}^{n} x_i w_i + B\right) \quad (1)$$

$$Y = \text{Sigmoid}\left(\sum_{i=0}^{n} x_i \beta_i + B\right) \quad (2)$$

For logistic regression, the inputs are generally normalized to compare coefficients ($\beta$). In the neural network, this is achieved by batch normalization (without center and scaling), normalizing the weights ($w$) after every activation. The weights can then be compared and used as an important measure. Since batch normalization is a batch-wise approximation, the learned weights can be multiplied with the standard deviation of the activations for a more accurate estimate. For Manhattan plots, the normalized, absolute weights between the gene layer and the output are used.

In larger networks, the relative importance is calculated by multiplying the weights for each possible path from the end node to the input SNPs. At each input, we obtain then a value denoting its contribution. These values are then aggregated (summed) according to the groups of the subsequent layers to get the importance estimate for each node in the network.

It is important to note that in this work (relative) importance is more similar to effect size or odds ratio than to statistical significance. The weights represent the direction and effect a gene or pathway has on the outcome of the network.

**Implementation**. Technically, the computational performance of the implemented Keras/Tensorfow[42,43] layer should be on par or an improvement over similar layers. It is implemented using sparse matrix multiplication, making it faster than the slice-wise locallyconnected1D layer and more memory efficient than dense matrix multiplication. The layer is friendly to use, with only one extra input compared to a normal dense layer. This extra input, the sparse connectivity matrix, is made with prior knowledge and describes how neurons are connected between layers.

The networks behave similar to normal fully connected artificial neural networks but are pruned by removing irrelevant connections:

$$Y = \text{Activation}\left(\sum_{i=0}^{n} x_i w_i + B\right) \quad (3)$$

With $w$ as a sparse matrix with learnable weights, initialized with a sparse connectivity matrix defining connections.

The GenNet framework is available as a command line tool and includes functionalities to convert data, to create, train, and evaluate and interpret neural networks.

**Baseline**. As a baseline method, LASSO logistic regression was implemented in Tensorflow by using a dense layer of a single neuron with a sigmoid activation function and L1 regularization on weights. Polygenic Risk Scores (PRS) were not used since we used only variants from the exome.

**Upper bound of classification accuracy for predicting phenotypes**. In contrast to the application of deep learning for image analysis, where the performance is not limited, in application for genetic analysis, we are limited by the information in DNA relevant for trait or disease, i.e., by the heritability. Therefore, we cannot expect the performance of the neural network above a certain value.

Population characteristics can be used to calculate the upper bound of performance for a classifier for any trait. This can be done by creating a confusion matrix. The accuracy between true and false positives for a perfect classifier, based solely on genetic inputs, is given by the concordance rate between monozygotic twins. It is impossible to predict better based solely on genetic code than the rate a trait occurs in people with virtually the same genetic code. The chance of misclassifying a control should be better than the prevalence, which is often close to zero for most diseases. Creating a confusion matrix can give insights in the upper bound for accuracy, sensitivity, and specificity in the dataset. An example for schizophrenia in our dataset can be found in Supplementary Method 6.

**Reporting summary**. Further information on research design is available in the Nature Research Reporting Summary linked to this article.

## Data availability

Code to run and generate data for the simulations are publicly available on GitHub. The genetic and phenotypic UK Biobank data are available upon application to the UK Biobank (https://www.ukbiobank.ac.uk/). Access to the Sweden-Schizophrenia Exome Sequencing study can be requested on dbGaP (https://www.ncbi.nlm.nih.gov/gap/) (dbGaP phs000473.v2.p2). All trained networks are available on https://github.com/ArnovanHilten/GenNet_Zoo/.

## Code availability

GenNet is an open-source framework usable from command line. GenNet and its tutorials, including how to build new layers and networks from prior knowledge, can be found on: https://github.com/arnovanhilten/GenNet/ and Zenodo[44].

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

## Acknowledgements

We would like to thank Marloes Arts (University of Copenhagen) for her advice. This work was funded by the Dutch Technology Foundation (STW) through the 2005 Simon Steven Meester grant 2015 to W.J. Niessen. This research has been conducted using the UK Biobank Resource under Application Number 23509. This work was carried out on the Dutch national e-infrastructure with the support of SURF Cooperative (application number 17610). The Rotterdam study is supported by the Netherlands Organization for Scientific Research (NWO, 91203014, 175.010.2005.011, 91103012).

## Author contributions

A.H. and G.R. conceived and designed the method. A.H. performed experiments and implemented the method. G.R. and W.N. supervised the work. M.A.I., C.K., H.A., M.K. and S.A.K. provided or gave access to data. A.H., G.R., W.N., M.K., M.A.I. and S.A.K, wrote, revised, and approved the paper.

## Competing interests

W.N. is co-founder and shareholder of Quantib BV. Other authors declare no competing interests.
