## [Transparent Peer Review File · Communications Biology]

Reviewers' comments:

Reviewer #1 (Remarks to the Author):

In this work the authors propose a deep learning framework for predicting phenotypes from genetic variants. Deep learning has been successfully implemented in areas such computer vision or robotics and is extremely useful when large amounts of data are available. Genomics could certainly benefit from deep learning, since genomic studies are driving into "big data" with an increasing pace. However, careful consideration is needed when dealing with the idiosyncrasy of genomic data. GWAS data usually include several thousands of samples and millions of variants (feature) and although deep learning could be proved useful in genetic variation data, sparsity of these data is a major drawback in applying deep learning. The second major drawback is that deep learning algorithms are often uninterpretable because of their complexity and in many genomic applications, researchers are more interested in the biological mechanisms revealed by the predictive model rather than the prediction accuracy itself. The most important step in building an effective deep learning model is to first curate an appropriate training dataset and to select a suitable evaluation metric. The training set should be designed appropriately balanced for confounders that would detrimentally affect performance when applied to external validation datasets. This is aka the generalization problem in machine learning and is the most important aspect to watch for when designing and applying any machine learning method.

To overcome the infeasible amount of computational time and memory that would need to train deep learning networks from genetic data due to data sparsity the authors propose to use information from annotated biological sources such as NCBI RefSeq, KEGG and single RNA gene expression datasets to define possible connections. The problem with depending so much on prior knowledge significantly eliminated knowledge discovery. If a variant has not been associated to a corresponding gene in the past, it will never be included in the model since it will have been discarded a priori.

Many important details are missing from the paper. For example, details regarding the type of deep learning architecture they are using, which hyperparameters they optimize, but most importantly how they optimize them. Tuning the hyper-parameters of an algorithm can make a world of a difference in predictive performance. Moreover, the authors do not describe their performance estimation protocol. Although they mention splitting their data into training, validation and test set, they do not give any details on how they compute performance. For example, when one selects the set of hyper-parameters that seem to provide the best estimated performance, this estimation is overestimated.

Overall, this is a very interesting work and topic in the field of machine learning and genomics. However, more details on the methodology and on the performance estimation protocol are needed to judge methodological correctness.

Reviewer #2 (Remarks to the Author):

This paper described a very interesting approach to phenotype prediction based on neural network models. The main innovation of this approach was to use biological knowledge to construct sparsely connected neural networks. This reduced the number of parameters that need to be learned from the data. The predictive performance of the proposed models exceeded that of LASSO. The weights in the neural networks showed the important genes and pathways for the phenotype prediction. I think this paper is acceptable for publication in this journal after major revisions to addressing the following comments:

In the Introduction section:

1) There are many algorithms developed to estimate polygenic risk scores (PRS). While LASSO

benchmarked here is reasonable, the authors should also compare with some other commonly used PRS algorithms, such as LDpred, pruning and thresholding, and BLUP. An example study is Khera et al 2018.

2) On line 33, the authors stated that "Deep learning is rarely used in population genomics". But there have been quite a few studies that explored this, including Ho et al 2019, Badre et al 2020, Bellot et al 2018, and Vivian-Griffiths et al 2019. The authors should benchmark their approach against general neural network models demonstrated in those previous studies and discuss the advantages of their new approach. The author should provide results to show how much improvement was obtained from the incorporation of biological knowledge (instead of just using neural networks). For example, a neural network model with random connections could be used as a baseline.

3) Biological knowledge has been used to help polygenic risk score estimations, including Amariuta et al 2020, Darst et al 2017, and Hari Dass et al 2019. Although these studies did not use neural networks, the authors should give a better coverage of the prior work in the Introduction

In the Results section:

4) The author should add a new performance benchmarking table which has all the studies in the rows of Table 1 and the performance metrics in the columns (e.g. test AUC) for each baseline algorithm requested above, such as LDpred, BLUP, densely connected neural network, sparsely connected neural network with random connection, etc. This will provide a quantitative assessment of the new approach over those existing approaches. This table is basically an expansion of their Supplementary 2.1 with more baseline models.

5) The authors should release their neural networks developed in this study, which is a common data sharing requirement in publications. The model parameters are not generally restricted by the genomics data protection policies of those databases, since they are just summary statistics over a cohort. Because the hidden nodes of these networks have biological meanings of pathways and genes, functional annotation of the hidden nodes should be provided along with the network. This will give the readers some concrete examples of networks from this study.

In the Methods section:

6) The comparison of Gene networks, pathway networks, and expression network in Supplementary 2 is very interesting. The author should provide more details about how these networks were constructed beyond the figure captions. For example, for each of these networks, how the data from KEGG, GTEx, etc were processed to determine which connections were incorporated into the neural networks. This would include all the thresholds used to filter the data. The architectures of these neural networks should also be specified in more details, such as the number of hidden neurons in each layer and the number of connections between the hidden layers. Basically, the authors could significantly expand the Prior knowledge section in lines 303~313 of the online methods with more technical details, since this was the core of this study.

7) I couldn't find any precise description of the training protocol for the neural network. Could the authors be more precise about the parameter tuning methods, the optimizer choice? Could the authors also specify if any additional hyperparameter were employed?

8) The calculation about the performance upper bound in Supplementary 7 was interesting. What is the reference for concordance rate = 0.5? It is not clear whether this was a made-up number for theoretical calculation or a real number from a biological study. Can the author do this for all the phenotypes with the experimentally determined concordance rates?

Reviewer #3 (Remarks to the Author):

The authors present a ML-powered framework for phenotype prediction that uses genetic information from public databases. By shaping the network itself using prior biological knowledge, the network weights become interpretable since nodes represent biological units and connections represent functional relationships. This makes the work presented here highly relevant and useful to a broad audience.

While the work is original and well founded in this manuscript, the reliance on existing gene and pathway annotations may limit the applicability and generalization of the framework since previously unknown interactions and functions will not be available to shape the neural network, leading to under-informed predictions. The authors briefly touch on this factor in the *Discussion* section of the paper. However, a deeper discussion on the impacts of missing annotations would clarify how applicable and resilient the method is.

For example, using data from the human studies that were used to validate the method (Table 1), how much of the relevant annotated data from public databases could be **deleted** while still achieving reasonable predictive value? I believe this is a more direct assessment of reliability than using synthetic data, which is much simpler.

The fact that the method was implemented in a broadly used infrastructure (as a TensorFlow layer), and distributed in an open-source, version-controlled repository also greatly enhances its reach and usability.

The method does achieve good results when challenged with data from human studies, and provides interesting insight into well-known and poorly-understood diseases. With the increasing number of public datasets at genome and metabolic network levels, tools such as the one described here will be of great help to discover and understand interactions between multiple biological factors that control disease onset and outcome.

Response to Reviewers

Reviewer 1

- 1) For example, details regarding the type of deep learning architecture they are using, which hyperparameters they optimize, but most importantly how they optimize them. Tuning the hyper-parameters of an algorithm can make a world of a difference in predictive performance. Moreover, the authors do not describe their performance estimation protocol. Although they mention splitting their data into training, validation and test set, they do not give any details on how they compute performance. For example, when one selects the set of hyper-parameters that seem to provide the best estimated performance, this estimation is overestimated.

We agree that in the original manuscript some information on the deep learning architecture, hyperparameters and optimization procedure was not provided and we would like to thank the reviewer for pointing this out. We have added Supplementary Material 2 with more details about the deep learning architectures such as the number of nodes and connections for all employed networks. In the Online Methods, neural network architecture, we have added missing information about the optimization procedure, hyperparameters and neural network architecture (line 330 to 357 in the revised manuscript).

For the reviewer's convenience lines 330 to 357 of the section neural network architecture are provided below:

However, the use of prior knowledge also restricts the network layout. The shape of the network, i.e., the number of neurons and the number of layers is determined by the prior knowledge embedded in the network (characteristics of all architectures can be found in Supplementary Material 2). Additionally, for a fair interpretation of the weights, each layer is preceded by batch normalization (without scaling and centering) resulting in an input for each layer with zero mean and unit standard deviation. For all classification task, a sigmoid function was used as a final activation function.

Operating within these restrictions, we optimized the following hyperparameters: the activation function, optimizer and the L1 kernel regularization penalty. In the first set of experiments to determine the activation function, we found that the hyperbolic tangent activation function consistently led to a higher area under the curve in the validation set than the Relu and PreLu activation functions for schizophrenia, hair color and eye color. Using the hyperbolic tangent activation function, we optimized the remaining hyperparameters. For each phenotype we tested different combinations of optimizers with different learning rates and L1 kernel regularization penalties. We explored combinations of the Adadelata optimizer (with learning rate of 1) or Adam optimizer (learning rates of 0.01, 0.001, 0.0001) with kernel L1 regularization penalties of 0.1, 0.001, 0.001, 0.00001 and 0 (no penalty). The L1 regularization penalty penalizes the network for the total sum of weights, similar to LASSO regression. A higher penalty will reduce the number of variants the network will use for prediction and is therefore related to the polygenicity of the phenotype.

All models were trained on a single GPU with 11 GB memory (Nvidia GeForce GTX 1080 Ti) using a batch size of 64. The networks were optimized using the ADAM or Adadelta optimizers, using weighted binary cross entropy with weights proportional to the imbalance of the classes. For regression tasks, mean squared error was used as a loss function in combination with ReLu activations (Supplementary Material 9). Training was stopped after not improving on the validation loss for 10 epochs (all models converged within five days). For each phenotype, the configuration of hyperparameters with the best performance in the validation set was used for a final evaluation in the test set.

Since we trained multiple networks for each phenotype, we could not extensively optimize all networks individually for each phenotype. We are convinced that with a more fine-grained hyperparameter search and by experimenting with more regularizers (e.g., dropout, reduce learning rate on plateau, or with activity regularizers) the performance of the neural networks can be further improved.

We would like to stress that we strictly optimized on the validation set and only after we finished optimizing, we evaluated the network once on the test set. For an indication of the performance for suboptimal hyperparameters we refer to Supplementary Table 7.1 where we trained the network with default parameters.

- 2) Overall, this is a very interesting work and topic in the field of machine learning and genomics. However, more details on the methodology and on the performance estimation protocol are needed to judge methodological correctness.

We would like to thank the reviewer for his/her kind words and interest in our work and we hope that the aforementioned changes provide more insight into the methodology and performance estimation. We hope that these additions provide the necessary details to judge methodological correctness.

Reviewer 2

- 1) There are many algorithms developed to estimate polygenic risk scores (PRS). While LASSO benchmarked here is reasonable, the authors should also compare with some other commonly used PRS algorithms, such as LDpred, pruning and thresholding, and BLUP. An example study is Khera et al 2018.

Based on the reviewers' comments 1, 2 and 4, we decided to add randomly connected neural network as an additional benchmark to the manuscript (Supplementary Material 7 and shown underneath the next comment in this document).

LASSO was used to benchmark the algorithm because we could train this with the same sample size and input variants as the neural networks in the GenNet framework. Any difference in performance could thus be attributed to the differences in the models. We have considered including polygenic risk scores as additional benchmarks but polygenic risk scores based on GWAS summary statistics require substantially larger sample size

and use different input variants (we focused on the exome, whereas polygenic risk scores are calculated using the whole genotype array). Therefore, calculating polygenic risk scores with the same input data as used in the experiments with the GenNet framework would result in an unfair comparison. We therefore think that LASSO regression represents a fairer comparison of PRS, which were calculated and applied within the same dataset as GenNet.

- 2) On line 33, the authors stated that "Deep learning is rarely used in population genomics". But there have been quite a few studies that explored this, including Ho et al 2019, Badre et al 2020, Bellot et al 2018, and Vivian-Griffiths et al 2019. The authors should benchmark their approach against general neural network models demonstrated in those previous studies and discuss the advantages of their new approach. The author should provide results to show how much improvement was obtained from the incorporation of biological knowledge (instead of just using neural networks). For example, a neural network model with random connections could be used as a baseline.

We agree with the reviewer that the statement on line 22 was too strongly phrased and this statement has been changed to:

Applying deep learning in population genomics is challenging because of computational issues and lack of interpretable models.

The algorithms referenced in Ho et al. (2019)⁴ and the convolutional networks of Bellot et al. (2018)⁵ and Badre et al. (2020)⁶ are indeed interesting examples of deep learning in population genetics. The main difference between the proposed method and these methods is that these methods rely on summary statistics to pre-select variants for training, whereas the networks used in the manuscript are not phenotype specific and did not pre-select SNPs. The work of Badre et al. (2020) is most comparable since they did also test a network for p-value threshold 1. In their work, they used a model with subsequently 1000, 250 and 50 nodes and 1 output node. Including all 528,620 SNPs in their model required a peak memory of 66.6 GB. If the required peak memory scales linearly, we could expect more than 900 GB GPU RAM for our 6.9 million inputs. Training such a network would exceed the computational resources we, and many others, have at our disposal. Moreover, training a network with more than 6.9 billion parameters would most likely also be impossible with our current sample size.

We would like to thank the reviewer for the proposed experiment with a randomly connected neural network. We think that the manuscript greatly benefitted from adding such an experiment to investigate the additional benefit of prior knowledge in terms of performance. The reviewer can find these experiments in Supplementary Material 7. The main advantage of incorporating prior knowledge is that we obtain an interpretable neural network. In Supplementary Material 7 we test if the networks with prior knowledge perform better than randomly connected neural networks without prior knowledge.

For the reviewer's convenience we show the relevant section below:

Supplementary Material 7. Does prior knowledge improve performance?

Embedding prior knowledge allows us to interpret the weights in the neural networks. One could speculate that embedding prior knowledge could also help in guiding training, resulting in better converged networks with improved performance over networks without prior knowledge.

This experiment is designed to test the hypothesis: ‘Embedding prior knowledge (gene annotations) in the neural network results in a network with a better performance than an equivalent network without prior knowledge’.

In this experiment, we used GenNet networks identical to the network used in the experiments of Supplementary Table 2.1 with gene annotations as prior knowledge embedded in the network.

The randomly connected networks are obtained by randomly shuffling the connectivity matrix in the horizontal direction. In the resulting network, all SNPs are randomly connected to nodes in the next layer (formerly known as the gene layer, now uninterpretable). All SNPs are thus connected to a random node and these nodes are connected to the output. Aside from this, the networks are equal in all aspects, they have the same number of trainable parameters (see Supplementary Material 2.1 Gene network) and all networks are trained with GenNet’s default hyperparameters (Adam with learning rate of 0.01 and L1 penalty of 0.01). We trained ten differently randomly connected networks and compared those to an equal number of GenNet gene networks for the Rotterdam Study and Sweden Schizophrenia. Due to limitations in resources, number of phenotypes and time constraints we decided to train six network per phenotype in the UK biobank (three shuffled and three regular). In total 112 networks were trained for this experiment.

Inspecting table Supplementary Table 7 shows that embedding prior knowledge in the neural network architecture does not lead to significantly better performance for all phenotypes. The results are inconclusive. For example, for red hair color we observe a non-significant improvement but this is not maintained for predicting black or blond hair color. The randomly connected network performs significantly better than a network with prior knowledge for schizophrenia but we find the opposite for predicting blue eye color in the Rotterdam color, GenNet significantly outperforms a randomly connected network. Thus, in general we cannot conclude that prior knowledge neither improves or deteriorates the performance.

Dataset	Trait	Subjects &	Number	AUC	AUC	AUC	AUC	P-value
(type)		phenotype	of runs	randomly	randomly	GenNet	GenNet (test)	Difference
		Class I	Class II	connected	connected	(val)		AUC
				network	network			(test)
				(val)	(test)			

Rotterdam (genotype array)	Eye color	4041 Blue	2250 Other	10	0.754±0.007	0.762±0.007	0.761±0.004	0.771 ± 0.002	6.54 × 10 ⁻³
UK Biobank (exome)	Hair color	4501 Blond	4518 Other	3	0.658±0.004	0.652±0.008	0.623±0.024	0.623±0.020	0.250
		15684 Dark brown	15918 Other	3	0.615±0.011	0.624±0.008	0.608±0.010	0.612±0.006	0.359
		1734 Red	1727 Other	3	0.837±0.021	0.834±0.017	0.907±0.018	0.900±0.022	0.084
		16208 Light brown	16029 Other	3	0.601±0.002	0.605±0.003	0.582±0.005	0.593±0.004	0.089
		3762 Black	3753 Other	3	0.831±0.003	0.818±0.004	0.820±0.010	0.800±0.011	0.068
	Atrial fibrillation	192 Cases	194 Controls	3	0.513±0.047	0.483±0.063	0.466±0.048	0.554±0.032	0.331
	Coronary Artery Disease	1563 Cases	1600 Controls	3	0.537±0.009	0.547±0.008	0.522±0.029	0.526±0.007	0.151
	Diabetes	2557 Cases	2555 Controls	3	0.544±0.003	0.545±0.002	0.527±0.046	0.524±0.020	0.201
	Dementia	139 Cases	142 Controls	3	0.466±0.033	0.492±0.069	0.531±0.069	0.464±0.083	0.677
	Allergies	10242 Cases	10187 Controls	3	0.522±0.01	0.513±0.004	0.489±0.007	0.505±0.005	0.283
	Breast cancer	1070 Cases	1082 Controls	3	0.529±0.017	0.525±0.015	0.529±0.012	0.539±0.020	0.618
	Asthma	4229 Cases	4214 Controls	3	0.534±0.009	0.531±0.005	0.507±0.010	0.51±0.020	0.230

Sweden (exome)	Schizophre nia	4969 Cases	6245 Controls	10	0.755±0.01	0.755±0.006	0.740±0.004	0.737±0.006	2.50 × 10 ⁻⁴
-------------------	-------------------	---------------	------------------	----	------------	--------------------	-------------	-------------	-------------------------

Supplementary Table 7. Overview of the experiments to determine if prior knowledge, aside from making the network interpretable, also improves performance. Per phenotype, the mean and standard deviations of the AUC over three runs for the UK biobank and ten runs for the other two datasets are shown. Significant better performance is emphasized.

- 3) Biological knowledge has been used to help polygenic risk score estimations, including Amariuta et al 2020, Darst et al 2017, and Hari Dass et al 2019. Although these studies did not use neural networks, the authors should give a better coverage of the prior work in the Introduction

We would like to thank the reviewer for this suggestion and the relevant references. We have added the following sentence in the introduction (line 61-63)

Additionally, it has been shown that embedding biological knowledge from these databases in polygenic risk scores can improve interpretation, trans-ancestry portability and genetic risk prediction.⁷⁻⁹

In the Results section:

- 4) The author should add a new performance benchmarking table which has all the studies in the rows of Table 1 and the performance metrics in the columns (e.g., test AUC) for each baseline algorithm requested above, such as LDpred, BLUP, densely connected neural network, sparsely connected neural network with random connection, etc. This will provide a quantitative assessment of the new approach over those existing approaches. This table is basically an expansion of their Supplementary Material 2.1 with more baseline models.

We would like to thank the reviewer for this excellent suggestion. We do agree that more quantitative assessment of the new approach over existing approaches is valuable. Unfortunately, proper densely connected networks are too memory intensive to use with the millions of variants in our input. We have added a quantitative assessment of GenNet versus a sparsely connected neural network with random connections in Supplementary Material 7.

However, as we mentioned in the response on comment 1) we do not think that LDpred, BLUP and other PRS approaches are methods that can be fairly compared to GenNet in the experiments we performed in this study. These methods are designed for genotype array data and use GWAS summary statistics, data derived from far larger sample sizes than used in our analysis. However, with new techniques like distributed learning (while guarantying safety of the data and privacy concerns) such sample sizes might be in reach for deep learning methods in the not-so-distant future.

- 5) The authors should release their neural networks developed in this study, which is a common data sharing requirement in publications. The model parameters are not generally restricted by the genomics data protection policies of those databases, since they are just summary statistics over a cohort. Because the hidden nodes of these networks have biological meanings of pathways and genes, functional annotation of the hidden nodes should be provided along with the network. This will give the readers some concrete examples of networks from this study.

We completely agree with the reviewer and have made most trained neural network architectures and annotations available (https://github.com/ArnovanHilten/GenNet_Zoo). All networks in the main table, Supplementary Materials 3, 4 and 5 are now publicly available.

The model parameters are indeed quite similar to summary statistics and we hope that one day the now publicly available networks will be as valuable as summary statistics have been in advancing the field.

In the Methods section:

- 6) The comparison of Gene networks, pathway networks, and expression networks in Supplementary Material 2 is very interesting. The author should provide more details about how these networks were constructed beyond the figure captions. For example, for each of these networks, how the data from KEGG, GTEx, etc were processed to determine which connections were incorporated into the neural networks. This would include all the thresholds used to filter the data. The architectures of these neural networks should also be specified in more detail, such as the number of hidden neurons in each layer and the number of connections between the hidden layers. Basically, the authors could significantly expand the Prior knowledge section in lines 312-320 of the online methods with more technical details, since this was the core of this study.

The incorporation of prior knowledge is indeed the core of this study and we acknowledge that we should have provided more details about the preprocessing and thresholds used for this data. We have rewritten the prior knowledge section and have added Supplementary Material 2 with more detailed information and links to the files used. Supplementary Material 2 also includes a table with the number of nodes and the number of hidden connections between each layer.

For the reviewer's convenience we have copied the relevant sections below:

Prior knowledge

All SNPs were annotated using Annovar³⁵. From these annotations a sparse connectivity matrix was created, connecting the SNPs to their corresponding genes. Connectivity matrices between SNPs, exons and genes were made using intron-exon annotations. A mapping between genes and pathways was made using GeneSCF³⁶ and the KEGG database⁶. Due to the hierarchical structure of the KEGG database no further preprocessing was necessary and the structure could be integrated in GenNet as it is. Expression based layers were created using publicly available group-wise t-score statistics by Finucane et al. (2018)²⁶. For each tissue, we connected the genes with the top 10% highest t-statistic to a tissue to ensures that nodes are uniquely defined by their connections and therefore interpretable. This threshold is in accordance with the work of

Finucane et al. (2018)²⁶, they showed that their results were stable for thresholds of 5%, 10% and 20% (URLs can be found in Supplementary Material 2).

Additional to the used layers, there are countless of different network layers possible. Any prior knowledge that groups data uniquely can be used to create layers in the GenNet framework.

Supplementary 2. GenNet architectures

2.1 All architectures: summary table

Dataset (type)	Number of input variants	Network type	Layer N of nodes (n connections)					Total number of trainable parameters	
Rotterdam (genotype array)	113,241 input variants	Gene network	Gene layer 16628 (129869)	Out 1 (16629)				146,498	
		Pathway network	Gene layer 16628 (129869)	Pathway 1 337 (21325)	Pathway2 44 (374)	Pathway 3 6 (50)	Out 1 (7)	151,625	
		GTEX expression networks	Gene layer 16356 (126716)	Tissue layer 53 (96327)	Out 1 (54)				223,097
		ImmGen expression networks	Gene layer 16628 (126716)	Cell layer 292 (428174)	Out 1 (293)				555,183
UK Biobank (exome)	6,986,636 input variants	Gene network	Gene layer 15827 (6661236)	Out 1 (15828)				6,677,064	
		Pathway network	Gene layer 15827 (6661236)	Pathway 1 337 (23550)	Pathway2 44 (374)	Pathway 6 (50)	Out 1 (7)	6,685,217	
		GTEX expression networks	Gene layer 21476 (6668279)	Tissue layer 53 (80249)	Out 1 (54)				6,748,582
		GTEX expression networks	Gene layer 21476 (6668279)	Tissue layer 53 (80249)	Out 1 (54)				6,748,582

		ImmGen expression networks	Gene layer 21476 (6668279)	Cell layer 292 (316342)	Out 1 (293)		6,984,914	
Sweden (exome variants)	1,288,701 input variants	Gene network	Gene layer 21390 (1310091)	Out 1 (21391)			1,331,482	
		Pathway network	Gene layer 21390 (1310091)	Pathway 1 330 (30851)	Pathway2 44 (432)	Pathway 3 6 (50)	Out 1 (7)	1,341,431
		GTEX brain expression networks	Gene layer 21390 (1310091)	Gene layer 23765 (1312466)	Tissue layer 13 (27253)	Out 1 (14)	1,339,733	
		GTEX expression networks	Gene layer 21390 (1310091)	Tissue layer 53 (109458)	Out 1 (54)		1,421,978	
		ImmGen expression networks	Gene layer 23765 (1312466)	Tissue layer 292 (468421)	Out 1 (293)		1,781,180	

Supplementary Table 1. Overview of the architectures used in this study. With the number of nodes in each layer and the number of weights/connections between brackets. The last column contains the number of trainable parameters for the architecture. The networks are phenotype independent, but do differ per dataset since each dataset contains different input variants.

2.2 Prior knowledge

Gene Layer: all SNPs are annotated using Annovar (see 2.3 and bibliography).² Using regular expression, all genes are filtered and SNPs without gene annotations are dropped. The complete pipeline can be found in:

https://github.com/ArnovanHilten/GenNet/blob/master/jupyter_notebooks/2_Define_connection_masks.ipynb

Pathway layer: All genes used in the gene layer are annotated using GeneSCF³ and connected to their subsequent pathways using the KEGG website (<https://www.genome.jp/kegg/pathway.html>).

ImmGen and GTEx: First we obtained the t-statistic matrices from Finucane et al. (2018)⁴. In their work Finucane et al computed for each gene a t-statistic for specific expression in the focal tissue. The 10% of genes with the highest t-statistic were assigned to the gene set corresponding to the focal tissue. The 10% threshold was chosen because it gave the most significant p-values in two of their datasets. In their evaluation of this parameter, they showed that their choice was valid, results did not change when using a 5% or 20% threshold (their Supplementary Figures 2a-c). Following their approach, we connected for each tissue the genes in the top 10% t-statistic tot that tissue.

2.3 URLs

Trained GenNet architectures deposit:

https://github.com/ArnovanHilten/GenNet_ModelZoo

Software

Annovar

<https://annovar.openbioinformatics.org/en/latest/> (free to use, sign-up required)

GeneSCF:

<https://github.com/genescf>

Data

ImmGen layer:

https://alkesgroup.broadinstitute.org/LDSCORE/LDSC_SEG_ldscores/tstats/ImmGen.tstat.tsv

GTEx expression layer:

https://alkesgroup.broadinstitute.org/LDSCORE/LDSC_SEG_ldscores/tstats/GTEx.tstat.tsv

Brain cell expression layer:

https://alkesgroup.broadinstitute.org/LDSCORE/LDSC_SEG_ldscores/tstats/GTEx_brain.tstat.tsv

scRNA-seq data (FUMA) layer:

https://github.com/Kyoko-wtnb/FUMA_scRNA_data

- 7) I couldn't find any precise description of the training protocol for the neural network. Could the authors be more precise about the parameter tuning methods, the optimizer choice? Could the authors also specify if any additional hyperparameter were employed?

We have added a description of the choices we made during optimization and information about the hyperparameters and training protocol to the neural network architecture section (line 330 to 357 in the revised manuscript).

For the reviewer's convenience lines 330 to 357 of the section neural network architecture are shown below:

However, the use of prior knowledge also restricts the network layout. The shape of the network, i.e., the number of neurons and the number of layers is determined by the prior knowledge embedded in the network (characteristics of all architectures can be found in Supplementary Material 2). Additionally, for a fair interpretation of the weights, each layer is preceded by batch normalization (without scaling and centering) resulting in an input for each layer with zero mean and unit standard deviation. For all classification task, a sigmoid function was used as a final activation function.

Operating within these restrictions, we optimized the following hyperparameters: the activation function, optimizer and the L1 kernel regularization penalty. In the first set of experiments to determine the activation function, we found that the hyperbolic tangent activation function consistently led to a higher area under the curve in the validation set than the Relu and PreLu activation functions for schizophrenia, hair color and eye color. Using the hyperbolic tangent activation function, we optimized the remaining hyperparameters. For each phenotype we tested different combinations of optimizers with different learning rates and L1 kernel regularization penalties. We explored combinations of the Adadelta optimizer (with learning rate of 1) or Adam optimizer (learning rates of 0.01, 0.001, 0.0001) with kernel L1 regularization penalties of 0.1, 0.001, 0.001, 0.00001 and 0 (no penalty). The L1 regularization penalty penalizes the network for the total sum of weights, similar to LASSO regression. A higher penalty will reduce the number of variants the network will use for prediction and is therefore related to the polygenicity of the phenotype.

All models were trained on a single GPU with 11 GB memory (Nvidia GeForce GTX 1080 Ti) using a batch size of 64. The networks were optimized using the ADAM or Adadelta optimizers, using weighted binary cross entropy with weights proportional to the imbalance of the classes. For regression tasks, mean squared error was used as a loss function in combination with ReLu activations (Supplementary Material 9). Training was stopped after not improving on the validation loss for 10 epochs (all models converged within five days). For each phenotype, the configuration of hyperparameters with the best performance in the validation set was used for a final evaluation in the test set.

- 8) The calculation about the performance upper bound in Supplementary 7 was interesting. What is the reference for concordance rate = 0.5? It is not clear whether this was a made-up number for theoretical calculation or a real number from a biological study. Can the author do this for all the phenotypes with the experimentally determined concordance rates?

We would like to thank the reviewer for his/her kind words. The experiment in Supplementary Material 7 (now 10) was used to provide an impression of the performance of the network compared to the maximally achievable performance. For schizophrenia, the concordance rate in literature for monozygotic twins varies from 0.41 to 0.65. We refined the calculations for schizophrenia and have added a table with calculations for each phenotype (with references). For the Reviewers convenience we have copied the relevant section below:

Supplementary 10.1 Overview of the upper bound classification accuracies in this study

Table with the upper bound accuracy according to the thought experiment in Supplementary 10.1. For some phenotypes such as hair color, the estimate might be less reliable due to migration (i.e., black hair color was natively not this prevalent in the UK). This methodological approach does not take in account migration, because of this the maximum accuracy might be underestimated for hair color.

Trait	Phenotype	Concordance	Prevalence	Cases	Controls	Max. Accuracy	Ref Conc.	Ref Prev.
Eye color	Blue	0.98	0.61	4041	2250	0.85	9	9
Hair color	Red	0.94	0.08	1734	1727	0.93	10	9
	Black	0.94	0.04*	3762	3753	0.95	11	11
	Brown	0.94	0.78	31892	31947	0.86	11	11
	Blond	0.94	0.41	4501	4518	0.76	11	11
Bipolar	Case	0.43	0.02	343	347	0.71	12	13
Atrial fibrillation	Case	0.22	0.03	192	194	0.60	14	15
CAD	Case	0.40	0.03	1563	1600	0.69	16	15
Dementia	Case	0.67	0.02	139	142	0.83	17	13
Asthma	Case	0.50	0.12	4229	4214	0.69	18	19
Diabetes type II	Case	0.87	0.05	2557	2555	0.91	20	21
Breast cancer	Case	0.28	0.01	1070	1082	0.64	22	23
Schizophrenia	Case	0.41	0.02	4969	6245	0.73	24	25, 13
Schizophrenia	Case	0.65	0.02	4969	6245	0.83	24	25, 13

*Supplementary Table 8 Overview of the estimated upper bound of the accuracy for the datasets used in this study. This table contains all relevant statistics used for this estimate (See supplementary 10). The monozygotic twin concordance and prevalence were obtained from literature. *This methodological approach does not take in account migration, because of this the upper bound accuracy might be underestimated.*

Reviewer #3 (Remarks to the Author):

The authors present a ML-powered framework for phenotype prediction that uses genetic information from public databases. By shaping the network, itself using prior biological knowledge, the network weights become interpretable since nodes represent biological units and connections represent functional relationships. This makes the work presented here highly relevant and useful to a broad audience.

We thank the reviewer for his/her positive comments on our work

- 1) While the work is original and well founded in this manuscript, the reliance on existing gene and pathway annotations may limit the applicability and generalization of the framework since previously unknown interactions and functions will not be available to shape the neural network, leading to under-informed predictions. The authors briefly touch on this factor in the Discussion section of the paper. However, a deeper discussion on the impacts of missing annotations would clarify how applicable and resilient the method is. For example, using data from the human studies that were used to validate the method (Table 1), how much of the relevant annotated data from public databases could be deleted while still achieving reasonable predictive value? I believe this is a more direct assessment of reliability than using synthetic data, which is much simpler.

Based on the reviewer's suggestion we have added Supplementary Material 8 in which we investigate the robustness of the method's predictive performance. In this experiment we plotted the number of deleted genes versus the AUC for schizophrenia and eye color (Figures 30 & 31). For the reviewer's convenience we have copied the relevant section below:

Supplementary Material 8. Deletion of predictive connections

In this experiment we evaluate the performance while deleting the connections to the most predictive features. We evaluated this for two widely different phenotypes, eye color, where *HERC2* and *OCA2* are the main contributors to the prediction of blue eye color and schizophrenia, a polygenic disease with numerous genes contributed to the prediction. As expected, the curves in Supplementary Figures 31 & 32 are different for the two phenotypes. The prediction for eye color deteriorates quickly, even by only deleting the connections to the *HERC2* gene the performance deteriorates, while the predictive performance of schizophrenia is relatively unaffected even if the connections to the top thousand predictive genes are deleted.

Supplementary Figure 31 & 32. Performance of the network while removing up to 20 000 connections. The genes are sorted by importance and deleted in this order, with connections to most important genes first deleted.

In the main text this is referred to as:

As a consequence, the predictive performance deteriorates slower for schizophrenia than for eye color prediction when connections to the most predictive genes are removed before training (see Supplementary 8).

- 2) The fact that the method was implemented in a broadly used infrastructure (as a TensorFlow layer), and distributed in an open-source, version-controlled repository also greatly enhances its reach and usability.

The method does achieve good results when challenged with data from human studies, and provides interesting insight into well-known and poorly-understood diseases. With the increasing number of public datasets at genome and metabolic network levels, tools such as the one described here will be of great help to discover and understand interactions between multiple biological factors that control disease onset and outcome.

We are grateful that the reviewer appreciates our work and sees potential for our method.

References response to authors:

1. Boix, C. A., James, B. T., Park, Y. P., Meuleman, W. & Kellis, M. Regulatory genomic circuitry of human disease loci by integrative epigenomics. *Nature* **590**, 1–8 (2021).
2. Chèneby, J. *et al.* ReMap 2020: A database of regulatory regions from an integrative analysis of Human and Arabidopsis DNA-binding sequencing experiments. *Nucleic Acids Res.* **48**, D180–D188 (2020).
3. Nasser, J. *et al.* Genome-wide enhancer maps link risk variants to disease genes. *Nature* (2021). doi:10.1038/s41586-021-03446-x
4. Ho, D. S. W., Schierding, W., Wake, M., Saffery, R. & O’Sullivan, J. Machine learning SNP based prediction for precision medicine. *Front. Genet.* **10**, 1–10 (2019).
5. Bellot, P., de los Campos, G. & Pérez-Enciso, M. Can deep learning improve genomic prediction of complex human traits? *Genetics* **210**, 809–819 (2018).
6. Badré, A., Zhang, L., Muchero, W., Reynolds, J. C. & Pan, C. Deep neural network improves the estimation of polygenic risk scores for breast cancer. *J. Hum. Genet.* **66**, 359–369 (2021).
7. Hari Dass, S. A. *et al.* A biologically-informed polygenic score identifies endophenotypes and clinical conditions associated with the insulin receptor function on specific brain regions. *EBioMedicine* **42**, 188–202 (2019).
8. Burcu F. Darst *et al.* Pathway-specific polygenic risk scores as predictors of β - amyloid deposition and cognitive function in a sample at increased risk for Alzheimer’s disease. *J Alzheimers Dis.* **176**, 139–148 (2017).
9. Amariuta, T. *et al.* Improving the trans-ancestry portability of polygenic risk scores by prioritizing variants in predicted cell-type-specific regulatory elements. *Nat. Genet.* **52**, 1346–1354 (2020).
10. Katsara, M. A. & Nothnagel, M. True colors: A literature review on the spatial distribution of eye and hair pigmentation. *Forensic Sci. Int. Genet.* **39**, 109–118 (2019).
11. Matheny, A. P. & Dolan, A. B. Changes in Eye colour during early childhood: Sex and genetic differences. *Ann. Hum. Biol.* **2**, 191–196 (1975).
12. Morgan, M. D. *et al.* The genetic architecture of hair colour in the UK population. *bioRxiv* (2018). doi:10.1101/320267
13. Kieseppä, T., Partonen, T., Haukka, J., Kaprio, J. & Lönnqvist, J. High concordance of bipolar I disorder in a nationwide sample of twins. *Am. J. Psychiatry* **161**, 1814–1821 (2004).
14. Prince, M. *et al.* No health without mental health. *Lancet* **370**, 859–877 (2007).
15. Christophersen, I. E. *et al.* Familial aggregation of atrial fibrillation: A study in danish twins. *Circ. Arrhythmia Electrophysiol.* **2**, 378–383 (2009).

16. Bhatnagar, P., Wickramasinghe, K., Wilkins, E. & Townsend, N. Trends in the epidemiology of cardiovascular disease in the UK. *Heart* **102**, 1945–1952 (2016).
17. Zdravkovic, S. *et al.* Heritability of death from coronary heart disease: A 36-year follow-up of 20 966 Swedish twins. *J. Intern. Med.* **252**, 247–254 (2002).
18. Gatz, M. *et al.* Heritability for Alzheimer's disease: The study of dementia in Swedish twins. *Journals Gerontol. - Ser. A Biol. Sci. Med. Sci.* **52**, 117–125 (1997).
19. Strachan, D. P., Wong, H. J. & Spector, T. D. Concordance and interrelationship of atopic diseases and markers of allergic sensitization among adult female twins. *J. Allergy Clin. Immunol.* **108**, 901–907 (2001).
20. Asthma statistics | British Lung Foundation. Available at: <https://statistics.blf.org.uk/asthma>. (Accessed: 12th May 2021)
21. Willemsen, G. *et al.* The Concordance and Heritability of Type 2 Diabetes in 34,166 Twin Pairs From International Twin Registers: The Discordant Twin (DISCOTWIN) Consortium. *Twin Res. Hum. Genet.* **18**, 762–771 (2015).
22. Bycroft, C. *et al.* The UK Biobank resource with deep phenotyping and genomic data. *Nature* **562**, 203–209 (2018).
23. Möller, S. *et al.* The heritability of breast cancer among women in the nordic twin study of cancer. *Cancer Epidemiol. Biomarkers Prev.* **25**, 145–150 (2016).
24. Forman, D. *et al.* Cancer prevalence in the UK: Results from the EUROPREVAL study. *Ann. Oncol.* **14**, 648–654 (2003).
25. Cardno, A. G. & Gottesman, I. I. Twin studies of schizophrenia: From bow-and-arrow concordances to star wars Mx and functional genomics. *Am. J. Med. Genet. - Semin. Med. Genet.* **97**, 12–17 (2000).
26. McGrath, J., Saha, S., Chant, D. & Welham, J. Schizophrenia: A concise overview of incidence, prevalence, and mortality. *Epidemiol. Rev.* **30**, 67–76 (2008).

REVIEWERS' COMMENTS:

Reviewer #1 (Remarks to the Author):

As general comments I note here the importance of hyper-parameter tuning and performance estimation in machine learning that very often lead to methodological errors. When one selects the set of hyper-parameters that seem to provide the best estimated performance a priori, this estimation is overestimated and should be adjusted. No adjustment is shown here. Also, performance metrics should be reported in confidence intervals. This is something that the authors here do not consider.

Reviewer #2 (Remarks to the Author):

This is an interesting study worth publishing in this journal. However, I still have some concerns about the content of the main text. The authors should be able to address these concerns just by revising the text without doing any new experiment.

Two general comments.

1) The authors have addressed my comments in their rebuttal letter. However, many of their responses are not reflected in the revised manuscript. The goal here is not just the rebuttal of the reviewers, but to improve the manuscript in case that other readers have similar questions and concerns. So, I would like to request the authors to incorporate their responses into the manuscript's main text.

2) The manuscript is still pretty short (only ~2800 words), so I don't understand why the authors pushed so much valuable results into the supplementary materials. I encourage the authors to pull some supplementary results to the main text.

A few specific comments to the authors' rebuttal letter.

1) "Based on the reviewers' comments 1, 2 and 4, we decided to add randomly connected neural network as an additional benchmark to the manuscript (Supplementary Material 7 and shown underneath the next comment in this document)."

Please describe and discuss this result in the main text. I can't even find Supplementary Material 7 referenced anywhere in the main text.

2) "We have considered including polygenic risk scores as additional benchmarks but polygenic risk scores based on GWAS summary statistics require substantially larger sample size and use different input variants (we focused on the exome, whereas polygenic risk scores are calculated using the whole genotype array). Therefore, calculating polygenic risk scores with the same input data as used in the experiments with the GenNet framework would result in an unfair comparison."

I disagree that it's unfair comparison. However, I respect the authors' opinion. As such, they should put their opinion in the main text for the record.

3) "The algorithms referenced in Ho et al. (2019)⁴ and the convolutional networks of Bellot et al. (2018)⁵ and Badre et al. (2020) are indeed interesting examples of deep learning in population genetics. The main difference between the proposed method and these methods is that these methods rely on summary statistics to pre-select variants for training, whereas the networks used in the manuscript are not phenotype specific and did not pre-select SNPs. The work of Badre et al. (2020) is most comparable since they did also test a network for p-value threshold 1. In their work, they used a model with subsequently 1000, 250 and 50 nodes and

1 output node. Including all 528,620 SNPs in their model required a peak memory of 66.6 GB. If the required peak memory scales linearly, we could expect more than 900 GB GPU RAM for our 6.9 million inputs.

Training such a network would exceed the computational resources we, and many others, have at our disposal. Moreover, training a network with more than 6.9 billion parameters would most likely also be impossible with our current sample size."

As above, this is fine. But please put your rationale here for not doing the comparison into the main text.

4) "We would like to thank the reviewer for the proposed experiment with a randomly connected neural network. We think that the manuscript greatly benefitted from adding such an experiment to investigate the additional benefit of prior knowledge in terms of performance. The reviewer can find these experiments in Supplementary main advantage of incorporating prior knowledge is that we obtain an interpretable neural Material 7. The network. In Supplementary Material 7 we test if the networks with prior knowledge perform better than randomly connected neural networks without prior knowledge."

I appreciate the authors for doing this experiment. However, the authors should describe the results and Supplementary Material 7 in the main text, instead of burying it in the Supplementary Materials. This is an important experiment, because one of the main premises of the study is to use the prior knowledge to improve the prediction performance. It's important to show the ambiguous results with improvement for some phenotypes, but not for some other phenotypes. This will not reduce the significance of the manuscript.

Reviewer #3 (Remarks to the Author):

I believe the authors have addressed all reviewer's comments to my satisfaction, and the manuscript can be accepted in the present form.

Response to Reviewers

Reviewer #1

- 1) As general comments I note here the importance of hyper-parameter tuning and performance estimation in machine learning that very often lead to methodological errors. When one selects the set of hyper-parameters that seem to provide the best estimated performance a priori, this estimation is overestimated and should be adjusted. No adjustment is shown here. Also, performance metrics should be reported in confidence intervals. This is something that the authors here do not consider.

We agree that obtaining a fair estimation of the performance is very important. In order to do this and avoid overfitting we split the data in a training, validation and test set, making sure that genetically related individuals are all in the training set. Then, we optimize the neural network using the training and validation set. Only the final model is evaluated on the holdout test set as is common practice for many deep learning applications^{1,2}. We chose this approach over cross-validation for confidence intervals to save resources and time. This allowed us to showcase the framework for more phenotypes.

We evaluated a limited number of hyperparameters (the activation function, optimizer and the L1 kernel regularization penalty) and for these hyperparameters we only tested a limited number of combinations due to computational limitations. For an impression of the performance of the neural networks with default hyperparameters we refer to Supplementary Table 7.

Reviewer #2

This is an interesting study worth publishing in this journal. However, I still have some concerns about the content of the main text. The authors should be able to address these concerns just by revising the text without doing any new experiment.

Two general comments.

- 1) The authors have addressed my comments in their rebuttal letter. However, many of their responses are not reflected in the revised manuscript. The goal here is not just the rebuttal of the reviewers, but to improve the manuscript in case that other readers have similar questions and concerns. So, I would like to request the authors to incorporate their responses into the manuscript's main text.

We are glad that we have addressed all comment in the rebuttal letter. We agree with the reviewer and we have added four new paragraphs in the introduction, result and discussion section that provide the information from the Supplementary and the responses to the reviewers'. We hope that with these changes the main text also addresses all the reviewer's concerns. The new paragraphs can be found on page 3, 7, 9, and 10. For the author's convenience they are also shown in the response to comment #3, #4 and #5.

- 2) The manuscript is still pretty short (only ~2800 words), so I don't understand why the authors pushed so much valuable results into the supplementary materials. I encourage the authors to pull some supplementary results to the main text.

We agree with the reviewer and have expanded the main text. As mentioned in the previous comment we think that with these additions the manuscript has improved significantly and we hope that the main text now contains all important findings. We have provided the highlighted version of the manuscript to make it easier to see the differences.

A few specific comments to the authors' rebuttal letter.

- 3) " Based on the reviewers' comments 1, 2 and 4, we decided to add randomly connected neural network as an additional benchmark to the manuscript (Supplementary Material 7 and shown underneath the next comment in this document)."

Please describe and discuss this result in the main text. I can't even find Supplementary Material 7 referenced anywhere in the main text.

We would like to thank the reviewer for pointing out the missing reference and the lack of information in the main text about the experiments with the randomly connected neural networks. We agree that information should have been provided in the main text. We now added two new paragraphs discussing this topic (page 7 and 10). Note that Supplementary 7 is now Supplementary Discussion 8.

For the reviewers' convenience we show the relevant sections below.

In the result section:

To evaluate if embedding prior knowledge also improves predictive performance, we compared networks built using gene annotations to networks with the same number of connections but randomly connected. For predicting blue eye color, the network build using gene annotations performed significantly better than a randomly connected neural network over ten runs ($p=6.5 \times 10^{-3}$). However, for predicting schizophrenia we found the opposite, a randomly connected neural network performed significantly better than networks embedded with prior knowledge ($p=2.5 \times 10^{-4}$). See Supplementary Discussion 8 for more information on all phenotypes.

In the discussion:

In a comparison between randomly connected neural networks and networks built using gene annotations we found that embedding gene annotations in the network architecture resulted in significantly better predictive performance for predicting eye color. However, randomly connected neural networks, with the same number of trainable parameters, performed significantly better for predicting schizophrenia. Based on these experiment using gene annotations, it seems that the optimal neural network architecture depends on the genetic architecture of the trait. But there are many factors that could influence the convergence and thus the predictive performance of a neural network aside from the genetic architecture of the trait. Among others, the quality of the available prior knowledge will be an important factors to consider.

- 4) "We have considered including polygenic risk scores as additional benchmarks but polygenic risk scores based on GWAS summary statistics require substantially larger sample size and use different input variants (we focused on the exome, whereas polygenic risk scores are calculated using the whole genotype array). Therefore,

calculating polygenic risk scores with the same input data as used in the experiments with the GenNet framework would result in an unfair comparison."

I disagree that it's unfair comparison. However, I respect the authors' opinion. As such, they should put their opinion in the main text for the record.

We agree that the response in the rebuttal letter should be reflected in the main text. As such, we have modified and added our response from the rebuttal letter to the discussion (page 9). For the reviewer's convenience we have copied the relevant section below:

For schizophrenia, polygenic risk scores from genome wide association studies obtained a similar AUC of 0.72 in an independent test cohort and a mean AUC of 0.70 in 40 leave-one-out GWAS analyses (AUC ranging from 0.49 to 0.81).³ However, a direct comparison between traditional polygenic risk scores and predictions given by neural networks from the GenNet framework would be unfair. The neural networks employed in this study use whole exome sequencing data or variants from the coding regions, while polygenic risk scores generally use all genotype data. Additionally, polygenic risk scores are built based on GWAS data with large sample sizes. For example for schizophrenia, the network was trained with roughly 4,696 cases and 6,969 controls while polygenic risk scores were derived using 32,838 cases and 44,357 controls.³

- 5) "The algorithms referenced in Ho et al. (2019)⁴ and the convolutional networks of Bellot et al. (2018)⁵ and Badre et al. (2020) are indeed interesting examples of deep learning in population genetics. The main difference between the proposed method and these methods is that these methods rely on summary statistics to pre-select variants for training, whereas the networks used in the manuscript are not phenotype specific and did not pre-select SNPs. The work of Badre et al. (2020) is most comparable since they did also test a network for p-value threshold 1. In their work, they used a model with subsequently 1000, 250 and 50 nodes and 1 output node. Including all 528,620 SNPs in their model required a peak memory of 66.6 GB. If the required peak memory scales linearly, we could expect more than 900 GB GPU RAM for our 6.9 million inputs. Training such a network would exceed the computational resources we, and many others, have at our disposal. Moreover, training a network with more than 6.9 billion parameters would most likely also be impossible with our current sample size."

As above, this is fine. But please put your rationale here for not doing the comparison into the main text.

We incorporated this information in the introduction (page 3) to make the reader aware of the work of Badre et al.⁴ and to point out the issues with fully connected neural networks when dealing with millions of input variants.

For the reviewer's convenience the relevant text is shown below:

Traditional fully connected neural networks have been successfully applied to predict genetic risk. Recently, Badre et al.⁴ employed a fully connected neural network for improving polygenic risk scores for breast cancer, training a neural network with up to 528,620 input variants. However, these networks are very memory-intensive and therefore often require pre-selecting SNPs using GWAS summary statistics. Applying these fully connected neural networks for millions of input variants would require infeasible amounts of computational time and memory.

- 6) "We would like to thank the reviewer for the proposed experiment with a randomly connected neural network. We think that the manuscript greatly benefitted from adding

such an experiment to investigate the additional benefit of prior knowledge in terms of performance. The reviewer can find these experiments in Supplementary main advantage of incorporating prior knowledge is that we obtain an interpretable neural Material 7. The network. In Supplementary Material 7 we test if the networks with prior knowledge perform better than randomly connected neural networks without prior knowledge."

I appreciate the authors for doing this experiment. However, the authors should describe the results and Supplementary Material 7 in the main text, instead of burying it in the Supplementary Materials. This is an important experiment, because one of the main premises of the study is to use the prior knowledge to improve the prediction performance. It's important to show the ambiguous results with improvement for some phenotypes, but not for some other phenotypes. This will not reduce the significance of the manuscript.

We thank the reviewer for this suggestion, and as we replied to comment #3, we have added two new sections discussing these results in the main text (page 7 and 10). In the result section we highlight the significant results of the experiment in Supplementary Discussion 8. In the discussion we emphasize that the genetic architecture of the trait and the quality of the prior knowledge are both important factors that should be considered when optimizing the network architecture for best predictive performance.

Reviewer #3:

I believe the authors have addressed all reviewer's comments to my satisfaction, and the manuscript can be accepted in the present form.

We are glad that the reviewer is satisfied with the manuscript and we appreciated all comments and suggestions which improved the manuscript.

References

1. Bishop, C. M. Model Assessment and Selection. in *Neural networks for pattern recognition*. 32–33 (Oxford university press, 1995).
2. Hastie, T., Tibshirani, R. & Friedman, J. The Curse of Dimensionality. in *The Elements of Statistical Learning* 219–223 (Springer, 2017).
3. Ripke, S. *et al.* Biological insights from 108 schizophrenia-associated genetic loci. *Nature* **511**, 421–427 (2014).
4. Badré, A., Zhang, L., Muchero, W., Reynolds, J. C. & Pan, C. Deep neural network improves the estimation of polygenic risk scores for breast cancer. *J. Hum. Genet.* **66**, 359–369 (2021).